# Evaluating Deep Unlearning in Large Language Models

## Abstract

Machine unlearning has emerged as an important component in developing safe and trustworthy models. Prior work on unlearning in LLMs has mostly considered unlearning tasks where a large corpus of copyrighted material or some specific training data are required to be removed. In this work, we consider the task of unlearning a fact from LLMs, which can be challenging as related facts can be deduced from each other, and investigate how well current unlearning methods for LLMs succeed at this task. Specifically, we formally propose a new setting of unlearning, *deep unlearning*, which considers fact unlearning under logical deductions between facts. We design a metric *recall*, to quantify the extent of deep unlearning. To enable us to systematically evaluate the extent of deep unlearning undistracted by other factors, we construct a synthetic dataset EDU-RELAT, which consists of a synthetic knowledge base of family relationships and biographies, together with a realistic logical rule set that connects them. We use this dataset to test four unlearning methods in four LLMs at different sizes. Our findings reveal that in the task of deep unlearning only a single fact, they either fail to properly unlearn with high recall, or end up unlearning many other irrelevant facts. Our results suggest that more targeted algorithms may have to developed for fact unlearning in LLMs. Our dataset and code are publicly available at : `https://anonymous.4open.science/r/deep_unlearning_anonymous-2C73`.

## 1 Introduction

Large language models (LLMs) of today are trained on massive amounts of uncurated data obtained from the internet. Machine unlearning in LLMs aims to remove specific pieces of data, concepts, or facts from these models in a more efficient way than retraining from scratch. These diverse definitions of unlearning (data, concept or fact unlearning) are tailored to different use cases. For instance, compliance with regulations such as the GDPR (Parliament & of the European Union, 2016) mandates the removal of a user's data. Similarly, unlearning (Eldan & Russinovich, 2023) can be used to address concerns that models retaining copyrighted material, for example the Harry Potter books, or offensive content.

In this paper, we consider the problem of unlearning facts from an LLM, which is important in scenarios with privacy requirements. Research has shown that LLMs can memorize personal and sensitive information (Carlini et al., 2021; Nasr et al., 2023), including relationships, work histories, and personal addresses. Such information can be readily accessed by LLM users, posing significant privacy risks and raising ethical concerns over uncontrolled exposure of private data. This motivates the need to unlearn facts.

Prior work have looked at fact unlearning problem, but the focus has been on removing the target fact itself. However, this can be superficial – LLMs not only know single facts in isolation, but many connected facts – and very often, the fact that has been unlearnt can be deduced from facts that are already known by the model. Thus, successful unlearning in this setting should also remove other facts that imply the fact to be unlearnt. As a concrete example, consider Figure 1. Here, the target fact *"Camila Flores's child is Wyatt Ross"* can be deduced from fact A *"Wyatt Ross's father is Xavier Ross"* and fact B *"Camila Flores's husband is Xavier Ross"*. If the LLM only unlearns

Figure 1: An example that unlearning only the target fact is insufficient. The successful extraction of *"Wyatt Ross's father is Xavier Ross"* and *"Camila Flores's husband is Xavier Ross"* can imply the target fact.

the target fact but retains A and B, this is insufficient as an adversary who extracts A and B from the LLM can deduce the target fact.

We consider a new setting for unlearning, which we call *deep unlearning*, and investigate to what extent current unlearning methods succeed at this setting. Deep unlearning is formulated by stating a set of facts and logical rules that connect the facts. The fact is deeply unlearnt if the target fact cannot be deduced from the retained facts in the LLM through the given logical rules. We further propose two metrics, recall and accuracy, for evaluating unlearning methods at deep unlearning. Recall measures to how well an unlearning method unlearns the relevant facts so that the target fact cannot be deduced; while accuracy measures what extent other irrelevant facts are retained by the unlearning process.

In order to have better control in evaluating deep unlearning, we construct a synthetic dataset EDU-RELAT as a benchmark. The dataset consists of two parts: a synthetic knowledge base and a realistic logical rule set. The knowledge base contains biographical information about a group of people (e.g., *"The birthyear of Sloane Lee is 1908"*), as well as family relationships (e.g., *"Wyatt Ross's father is Xavier Ross"*). The logical rules are realistic rules describing family relationships (e.g. (X, *husband*, Z)∧ (Y, *father*, Z) → (X, *child*, Y)). The unlearning method is tested on a subset of facts on family relationships together with the rules.

We then use our dataset to evaluate four common unlearning methods (Gradient Ascent, Negative Preference Optimization, Task Vector, and Who's Harry Potter) on four popular LLMs (Phi-1.5, GPT2-XL, Llama2-7b, Llama3-8b). We find that while these methods are good at unlearning the target fact itself without losing accuracy, they either fail to deeply unlearn with high recall or lose more than 20% irrelevant facts while deeply unlearning just single target fact. Additionally, it is found that the unlearning methods have better performance on larger LLMs; this validates the hypothesis that more inherent understanding of facts in larger LLMs helps with deep unlearning naturally.

This illustrates that the machine unlearning methods of today are largely insufficient for properly unlearning facts from LLMs; either they do not unlearn what is needed to deduce a target fact, or they over-unlearn and unlearn many extraneous facts. We hypothesize that this might be because the existing unlearning methods do not sufficiently account for the nature of facts and the reasoning capabilities of LLMs. We posit that future methods that unlearn facts suitably from LLMs should be aware of these enhanced capabilities.

## 2 PRELIMINARY

Given a set of objects $\mathcal{O}$ and relations $\mathcal{T}$, a fact $k$ is represened by the triplet $(o_1, r, o_2)$ of the relation $r$ and two objects $o_1, o_2$. For example, *"Camila Flores's child is Wyatt Ross"* can be represented in (*Camila Flores*, *child*, *Wyatt Ross*). The knowledge base $\mathcal{K} \subseteq \mathcal{O} \times \mathcal{T} \times \mathcal{O}$ is a set of facts.

Facts can be deduced from each other through reasoning; an example has been shown in Figure 1. A popular way to formulate the reasoning among facts is through a logical rule $R$, which is widely used to discover new knowledge (Galárraga et al., 2013; Yang et al., 2017; Xu et al., 2022; Cheng et al., 2023; Luo et al., 2023). The rule $R$ has the form of $B_1 \wedge \cdots \wedge B_n \to A$, where $B_1 \cdots, B_n$ and $A$ are atoms and each atom is a tuple $(X, r, Y)$ of logical variables $X, Y$ and a relation $r$. One example of rule is (X, *husband*, Z)∧ (Y, *father*, Z) → (X, *child*, Y). By substituting the objects in $\mathcal{O}$ to the logical variables in $B_1, \cdots, B_n, A$, facts on the left can together deduce the fact on the right.

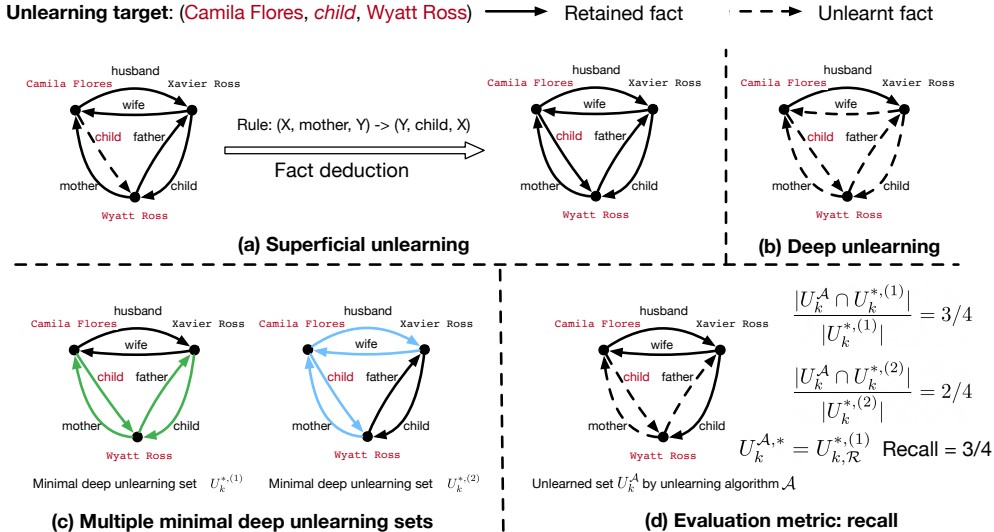

Figure 2: An illustration of *deep unlearning*. (a) an example of superficial unlearning; (b) an example of deep unlearning; (c) two different minimal deep unlearning sets for unlearning the same target fact; (d) the calculation of our proposed evaluation metric *recall*.

With a set of rules $\mathcal{R}$, a knowledge base $\mathcal{K}$ is *deductively closed* (Cheney et al., 2009; Cohen, 2016; Huang et al., 2021) with respect to $\mathcal{R}$, if there is no new fact can be deduced from $\mathcal{K}$ and $\mathcal{R}$. For example, suppose (*Camila Flores*, *husband*, *Xavier Ross*), (*Xavier Ross*, *child*, *Wyatt Ross*) $\in \mathcal{K}$ and (X, *husband*, Z)$\wedge$ (Y, *father*, Z) $\to$ (X, *child*, Y) is in the rule set $\mathcal{R}$. If $\mathcal{K}$ is deductively closed with respect to the rule set including the rule set $\mathcal{R}$, (*Camila Flores*, *child*, *Wyatt Ross*) must be in the knowledge base $\mathcal{K}$ too. The deductive closure is defined as follows.

**Definition 1** (Deductive closure). *The deductive closure of $\mathcal{K}$ with respective to the rule set $\mathcal{R}$, notated as $\Omega(\mathcal{K}, \mathcal{R})$, is the smallest set such that 1. $\mathcal{K} \subseteq \Omega(\mathcal{K}, \mathcal{R})$; 2. $\Omega(\mathcal{K}, \mathcal{R})$ is deductively closed with respect to a set of rules $\mathcal{R}$.*

## 3 DEEP UNLEARNING

Prior work in fact unlearning from LLMs focuses on simply unlearning the target fact in isolation, and not other facts that logically imply it. This might cause the LLM to forget only this one specific fact, but retain others that can be combined to deduce the fact in question. In this section, we introduce the new setting of unlearning, deep unlearning, which considers the logical deductions between facts. To assess the effectiveness of unlearning methods in this setting, we propose two evaluation metrics: recall and accuracy.

### 3.1 FACT DEEP UNLEARNING

Let $\mathcal{K}$ represent the knowledge base of the LLM prior to unlearning and let $U_k^{\mathcal{A}} \subseteq \mathcal{K}$ denote the set of facts that has been removed by any unlearning method $\mathcal{A}$ aimed at unlearning the target fact $k$. If method $\mathcal{A}$ deeply unlearns the fact $k$, it is expected that the fact $k$ should not be deduced from the retained facts $\mathcal{K} \backslash U_k^{\mathcal{A}}$ by the rule set $\mathcal{R}$, i.e. $k$ should not be in the deductive closure $\Omega(\mathcal{K} \backslash U_k^{\mathcal{A}}, \mathcal{R})$.

**Definition 2** (deep unlearning ). *The unlearning method $\mathcal{A}$ deeply unlearns the fact $k$ with respect to the rule set $\mathcal{R}$ if the fact $k$ does not belong in the deductive closure $\Omega(\mathcal{K} \backslash U_k^{\mathcal{A}}, \mathcal{R})$.*

We call the unlearning, which successfully unlearns the target fact but does not satisfy deep unlearning, as *superficial* unlearning; we show an example in Figure 2(a). Figure 2(b) shows an example of deep unlearning; from the only retained fact (*Camila Flores*, *husband*, *Xavier Ross*), the target fact cannot be deduced by any rules. We further notice that in the example of deep unlearning (Figure 2(b)), even if (*Xavier Ross*, *wife*, *Camila Flores*) is not unlearnt, it is still an example of deep

---

**Algorithm 1** DUS$(k, \mathcal{K}, \mathcal{R})$ – Random generation of the *Deep Unlearning Set*

---

**Input:** The target fact $k$, the knowledge base $\mathcal{K}$, the rule set $\mathcal{R}$.

1: $\hat{U}_k = \{k\}, T = \{k\}$
2: **while** $T \neq \emptyset$ **do**
3:    Uniformly randomly pick $k_{\mathrm{cur}}$ from $T$. $T = T \backslash \{k_{\mathrm{cur}}\}$
4:    Find all initializations of rules $\mathcal{I}_{k_{\mathrm{cur}}}$ that implies $k_{\mathrm{cur}}$ and denote the size $|\mathcal{I}_{k_{\mathrm{cur}}}|$ as $m_{k_{\mathrm{cur}}}$:

$$\mathcal{I}_{k_{\mathrm{cur}}} = \{I_j | \forall j \in [m_{k_{\mathrm{cur}}}], I_j = (k_1^j, \cdots, k_{n_j}^j, k_{\mathrm{cur}}) \in \Omega(\mathcal{K}, \mathcal{R}) \times \cdots \times \Omega(\mathcal{K}, \mathcal{R})$$

$$\text{is an initiation of the rule } B_1^j \wedge \cdots \wedge B_{n_j}^j \to A_j \in \mathcal{R}\}$$

5:    **for** $(k_1^j, \cdots, k_{n_j}^j, k_{\mathrm{cur}})$ in $\mathcal{I}_{k_{\mathrm{cur}}}$ where $\{k_1^j, \cdots, k_{n_j}^j\} \cap \hat{U}_k = \emptyset$ in a random order **do**
6:       Uniformly randomly pick $k^j$ from $\{k_1^j, \cdots, k_{n_j}^j\}$. $\hat{U}_k = \hat{U}_k \cup \{k^j\}, T = T \cup \{k^j\}$.
7:    **end for**
8: **end while**
9: **Output:** $U_k := \hat{U} \cap \mathcal{K}$.

---

unlearning. In practice, we would prefer the LLM that deeply unlearns the target fact but retains other facts as much as possible. Therefore, We define what *minimal* deep unlearning is.

**Definition 3** (Minimal deep unlearning ). *Given a fact $k$, the minimal deep unlearning set $U_k^*$ to unlearn the fact $k$ with respect to the rule set $\mathcal{R}$ should meet two requirements: 1. $k \notin \Omega(\mathcal{K}\backslash U_k^*, \mathcal{R})$, 2. $\forall U \subset U_k^*$, $k \in \Omega(\mathcal{K}\backslash U, \mathcal{R})$. Moreover, the unlearning method minimally deeply unlearns $k$ with respect to the rule set $\mathcal{R}$ if $U_k^{\mathcal{A}}$ is any minimal deep unlearning set.*

Note that minimal deep unlearning set needs not be unique. For example, Figure 2(c) shows two possible minimal deep unlearning sets for unlearning the same target fact.

## 3.2 EVALUATION METRIC

---

**Algorithm 2** RP$(k, \mathcal{K}, \mathcal{R}, U_k)$ – *R*andom *P*runing the deep unlearning set

---

**Input:** The target fact $k$, the knowledge base $\mathcal{K}$, the rule set $\mathcal{R}$, the deep unlearning set $U_k$

1: $C = \{\}, t = 0, U_k^* = U_k$.
2: **while** $C \neq \emptyset$ or $t = 0$ **do**
3:    $C = \{\}, t = t + 1$
4:    **for** $k_{\mathrm{cur}}$ in randomly shuffled $U_k^*$ **do**
5:       **if** $k \notin \Omega(\mathcal{K}\backslash(U_k^*\backslash\{k_{\mathrm{cur}}\}), \mathcal{R})$ **then**
6:          $C = C \cup \{k_{\mathrm{cur}}\}, U_k^* = U_k^*\backslash\{k_{\mathrm{cur}}\}$
7:       **end if**
8:    **end for**
9: **end while**
10: **Output:** $U_k^*$.

---

**Algorithm 3** MDUS$(k, \mathcal{K}, \mathcal{R}; N_{\mathrm{seed}})$ – Generating multiple *M*inimal *D*eep *U*nlearning *S*ets

---

**Input:** The target fact $k$, the knowledge base $\mathcal{K}$, the rule set $\mathcal{R}$, the number of seeds $N_{\mathrm{seed}}$.

1: $\hat{\mathcal{M}}_{k, \mathcal{R}, \mathcal{K}} = \{\}$.
2: **for** $n_{\mathrm{seed}} = 1, \cdots, N_{\mathrm{seed}}$ **do**
3:    $U_k$ =DUS$(k, \mathcal{K}, \mathcal{R})$. \\ Algorithm 1
4:    $U_k^*$=RP$(k, \mathcal{K}, \mathcal{R}, U_k)$. \\ Algorithm 2
5:    $\hat{\mathcal{M}}_{k, \mathcal{R}, \mathcal{K}} = \hat{\mathcal{M}}_{k, \mathcal{R}, \mathcal{K}} \cup \{U_k^*\}$.
6: **end for**
7: **Output:** $\hat{\mathcal{M}}_{k, \mathcal{R}, \mathcal{K}}$

---

We propose two evaluation metrics to evaluate an unlearning method $\mathcal{A}$: *recall* and *accuracy*. Recall is to measure the extent of deep unlearning of an unlearning method $\mathcal{A}$. It calculates the percentage of any minimal deep unlearning set that has been unlearnt by the method $\mathcal{A}$. Because the minimal deep unlearning set is not unique, the recall is defined w.r.t. the minimal deep unlearning set that $U_k^{\mathcal{A}}$ covers the most. Formally, let $\mathcal{M}_{k, \mathcal{R}, \mathcal{K}}$ denote the set of all minimal deep unlearning sets to unlearn $k$ (from the knowledge base $\mathcal{K}$ with respective to the rule set $\mathcal{R}$). The *recall* for a given unlearning method $\mathcal{A}$ to unlearn $k$ is defined as

$$\mathrm{Recall}(\mathcal{A}, k; \mathcal{K}, \mathcal{R}) = \max_{U_k^* \in \mathcal{M}_{k, \mathcal{R}, \mathcal{K}}} \frac{|U_k^{\mathcal{A}} \cap U_k^*|}{|U_k^*|}. \tag{1}$$

We also denote $U_k^{\mathcal{A},*} := \arg\max_{U_k^* \in \mathcal{M}_{k,\mathcal{R},\mathcal{K}}} \frac{|U_k^{\mathcal{A}} \cap U_k^*|}{|U_k^*|}$ the minimal deep unlearning set that $U_k^{\mathcal{A}}$ covers the most, which is used for calculating the recall. Figure 2(d) shows an example of calculating this recall. There are two minimal deep unlearning sets for unlearning the target fact and by definition $U_k^{\mathcal{A},*} = U_k^{*,(1)}$ is picked for the final recall value.

Subsequently, we can define the set of facts to measure the accuracy of the LLM. $U_k^{\mathcal{A},*}$, as previously defined, is the minimal deep unlearning set to calculate the recall. We calculate the accuracy among the knowledge base excluding this minimal deep unlearning set, $\mathcal{K} \backslash U_k^{\mathcal{A},*}$, for measuring the model utility:

$$\text{Accuracy}(\mathcal{A}, k; \mathcal{K}, \mathcal{R}) = \frac{|(\mathcal{K} \backslash U_k^{\mathcal{A},*}) \backslash U_k^{\mathcal{A}}|}{|\mathcal{K} \backslash U_k^{\mathcal{A},*}|}. \tag{2}$$

At the ideal situation where the unlearning method $\mathcal{A}$ exactly unlearns a deep unlearning set, both recall and accuracy are 1; otherwise, either the unlearning method does not deeply unlearn the target fact (recall$< 1$), or it unlearns extraneous facts (for unlearning the target fact $k$) (accuracy$< 1$).

**Approximation algorithm for calculating recall and accuracy.** The calculations of both recall and accuracy rely on an optimization problem

$$U_k^{\mathcal{A},*} := \arg\max_{U_k^* \in \mathcal{M}_{k,\mathcal{R},\mathcal{K}}} \frac{|U_k^{\mathcal{A}} \cap U_k^*|}{|U_k^*|},$$

where $\mathcal{M}_{k,\mathcal{R},\mathcal{K}}$ denote the set of all minimal deep unlearning sets to unlearn $k$ (from the knowledge base $\mathcal{K}$ with respective to the rule set $\mathcal{R}$). However, finding this exact $U_k^{\mathcal{A},*}$ in general can be NP-hard (Skiena, 2020). Alternatively, we propose Algorithm 3, which are able to find multiple minimal deep unlearning sets $\hat{\mathcal{M}}_{k,\mathcal{R},\mathcal{K}}$. With $\hat{\mathcal{M}}_{k,\mathcal{R},\mathcal{K}}$, it is efficient to find $\hat{U}_k^{\mathcal{A},*} := \arg\max_{U_k^* \in \hat{\mathcal{M}}_{k,\mathcal{R},\mathcal{K}}} \frac{|U_k^{\mathcal{A}} \cap U_k^*|}{|U_k^*|}$ and approximately calculate the recall and accuracy afterwards.

The idea in Algorithm 3 is to generate a single deep unlearning set with some randomness (line 3) and to repeat this generation process to attain multiple different deep unlearning sets (line 4); the proof that $\hat{M}_{k,\mathcal{R},\mathcal{K}}$ returned by Algorithm 3 is a collection of minimal deep unlearning sets is in Appendix A. There are two steps to find a single minimal deep unlearning set;

1. Find the deep unlearning set (Algorithm 1). We enumerate the rules and find all combinations of facts that can imply $k$ (line 4). For each combination, if no facts in this combination are in the returning set $U_k$, we randomly pick one fact from this combination and add it to the returning set $U_k$ (lines 5-7). Additionally, for the picked fact in any combination, we repeat the above process but for this fact recursively. This algorithm guarantees that $k \notin \Omega(\mathcal{K} \backslash U_k, \mathcal{R})$ and randomness from picking fact in each combination and the order for going through the combinations brings the diversity of the results.
2. Prune $U_k$ to a minimal deep unlearning set $U_k^*$ (Algorithm 2). We go through every fact $k_{\text{cur}}$ in $U_k$ one by one and check if $U_k \backslash \{k_{\text{cur}}\}$ from $\mathcal{K}$ is still a deep unlearning set. If yes, we can safely remove $k_{\text{cur}}$ from current $U_k$ and repeat this process until there is no $k_{\text{cur}} \in U_k$ that can be removed. The $U_k^*$ returned by this algorithm is guaranteed to be a minimal deep unlearning set, and the randomness in the order of checking $k_{\text{cur}} \in U_k$ brings the diversity of the results.

By running Algorithm 3 on the facts in the synthetic dataset introduced in the later section, we find that Algorithm 3 is capable of generating a diverse set of minimal deep unlearning sets. For more than half of the facts in our synthetic dataset, Algorithm 3 can return 6-17 different minimal deep unlearning sets. This demonstrates the effectiveness of Algorithm 3 and hence leads to a good approximation for computing the recall in Equation 1. Please check more details together with the example of minimal deep unlearning sets found by Algorithm 3 in Appendix C.

## 4 EDU-RELAT: EVALUATING DEEP UNLEARNING THROUGH RELATIONSHIPS

To systematically evaluate deep unlearning in LLMs, we need a dataset that is already in the LLMs and consists of multiple instances where one or more facts imply other facts by some realistic rules.

| Fact | Question | Answer |
|---|---|---|
| (Reid Perry, *father*, Richard Perry) | Who is Richard Perry to Reid Perry? | Father |
| (Richard Perry, *child*, Quentin Perry) | Who is Quentin Perry to Richard Perry? | Child |
| (Quinn Gray, *sister*, Rachel Gray) | Who is Rachel Gray to Quinn Gray? | Sister |
| (Sloane Lee, *birthyear*, 1908) | What is the birth year of Sloane Lee? | 1908 |
| (Sloane Lee, *birthplace*, Washington state) | What is the birthplace of Sloane Lee? | Washington state |
| (Sloane Lee, *job*, Banker) | What is the job of Sloane Lee? | Banker |

Table 1: Examples of synthetic facts in family relationships and biography.

| | |
|---|---|
| (B, *mother*, A) → (A, *child*, B) | (B, *father*, A) → (A, *child*, B) |
| (C, *mother*, A) ∧ (B, *brother*, C) → (A, *child*, B) | (C, *mother*, A) ∧ (B, *sister*, C) → (A, *child*, B) |
| (C, *father*, A) ∧ (B, *sister*, C) → (A, *child*, B) | (C, *father*, A) ∧ (B, *brother*, C) → (A, *child*, B) |
| (A, *child*, C) ∧ (B, *sister*, C) → (A, *child*, B) | (A, *child*, C) ∧ (B, *brother*, C) → (A, *child*, B) |
| (A, *child*, C) ∧ (B, *wife*, C) → (A, *child*, B) | (A, *child*, C) ∧ (B, *husband*, C) → (A, *child*, B) |

Table 2: Examples of rules in $\mathcal{R}$ that deduce the fact that has *child* as relation.

One plausible way of constructing such a dataset is to use real-world knowledge bases such as the triplets in Wikipedia dump[1]. However, we find that evaluating unlearning on real-world facts can be noisy due to two factors:

1. Partial Observation of the underlying LLM-Knowledge-Base: Reconstructing a real-world knowledge base from an existing public one only gives us a partial observation of the underlying knowledge base in the LLM, because public real-world knowledge bases are already incomplete and the process of checking if a fact is in an LLM is difficult at the engineering level as mentioned in Zhong et al. (2023). A partial observation of the underlying knowledge base in the LLM can falsely indicate the success of deep unlearning: it is possible that even post unlearning, some facts that deduce the unlearnt target are still retained, while the evaluation result indicates the success of this unlearning just due to the absences of the retained facts in the observed knowledge base.

2. Different underlying knowledge bases across LLMs: The underlying knowledge bases for different LLMs are different. Hence the same target fact can have different minimal unlearning sets, leading to different behavior of a given unlearning method across different LLMs. This makes it harder to make consistent conclusions for an unlearning method across LLMs (for example, any unlearning method is best for all LLMs).

Therefore, to have better control on the evaluation, we construct a synthetic dataset named EDU-RELAT for systematically evaluating deep unlearning through relationships in the family. We locate our synthetic dataset in a family network, which is a common scenario to study rule mining and knowledge discovery in the literature (Galárraga et al., 2013; Cheng et al., 2023; Luo et al., 2023). This synthetic dataset includes a synthetic knowledge base consisting of 400 family relationships and 300 biographical facts among 100 fictitious people, as well as a set of realistic logical rules, which are deductions among family relationships. Family relationships include *child*, *father*, *mother*, *husband*, *wife*, *brother*, *sister*, *aunt*, *uncle*, *nephew*, *niece*. Biographies include *birthyear*, *birthplace*, and *job*. Table 1 shows some examples of facts in family relationships and biographies, together with the question-answer pairs for checking whether this fact is in the LLM or not. Moreover, the rule set $\mathcal{R}$ has 48 rules, which are used to deduce the facts in family relationships. Table 2 shows all rules that can imply the fact that has *child* as the relationship.

We make several efforts to better mimic a knowledge base of real-world including:

- **Family network generation.** We recursively expand the network. Given a node (person), with a certain probability, we generate the parents, spouse, and children of this person. We control the whole family network in 4 generations. The number of children from any couple is sampled from a truncated geometric distribution between 1 and 4.
- **Name generation.** We collect two lists of first names for males and females separately and assign the first name to each person according to gender. As for the last name, each person's last name is the same as the father's if the father exists in the network. There is only one special case where the female's last name has a small probability of switching to her husband's.
- **Biography generation.** We have three biographical attributes, birth year, birthplace, and job:

---

[1]https://github.com/neelguha/simple-wikidata-db

- The birth years of people are aligned with their relationships. The birth year of any child is from a truncated Gaussian distribution given his/her mother's birth year. The difference in birth years of a couple is sampled from a reasonable distribution as well.
- The birthplace is the state in the United States. The child's birthplace is the same as the birthplace of the parent with a high chance, or sampled from the population distribution in the United States.
- The job list is collected from GPT4 for every ten years in 1900-2020. The job of a person is picked based on the birth year.

We believe these realistic considerations reduce the gap in evaluations between the unlearning task in our synthetic dataset and the real-world unlearning task. More statistics of this synthetic dataset are presented in Appendix B.

Within the synthetic dataset EDU-RELAT, we finetune pre-trained LLM on it for the next unlearning evaluation. The unlearning method is run with the finetuned LLM and tested on a subset of facts on family relationships together with the rules. The recall defined in Equation 1 is calculated with an approximation Algorithm 3; in addition, we measure the accuracy of the remaining facts in both family relationships and biographies to reflect the cost of unlearning.

## 5 EXPERIMENTS

In this section we investigate to what extent current unlearning methods succeed at deep unlearning. We release our dataset and code as a benchmark at : https://anonymous.4open.science/r/deep_unlearning_anonymous-2C73.

### 5.1 SET-UP

**Unlearning methods.** For each target LLM, we evaluate four common unlearning methods in the literature: *gradient ascent*, *negative preference optimization*, *task vector* and *who's harry potter*.

- *Gradient ascent* (GA; (Jang et al., 2022)) maximizes the loss on the target data, which is a reversed process of learning that does gradient *descent*. More optimization steps $T$ result in better unlearning but worse accuracy on extraneous facts.
- *Negative preference optimization* (NPO; (Zhang et al., 2024)) minimizes the difference between the likelihood of the target data $L(x_{\text{target}}; f_\theta)$ and the likelihood $L(x_{\text{target}}; f_{\text{original}})$ from the original model $f_{\text{original}}$, while not allowing the unlearnt model to diverge too much from the original model. The objective is defined as $\mathcal{L}(x_{\text{target}}, \theta) = -\frac{2}{\beta} \log \sigma \left( \beta \log \left( \frac{L(x_{\text{target}}; f_\theta)}{L(x_{\text{target}}; f_{\text{original}})} \right) \right)$. As suggested by the literature, the parameter $\beta$ that controls the degree of divergence between unlearnt and original models is set to $0.1$. Similar to gradient ascent, the step $T$ of optimizing $\mathcal{L}(x_{\text{target}}, \theta)$ is to control the trade-off between the unlearning performance and the model utility.
- *Task vector* (TV; (Ilharco et al., 2023)) first finetunes the original model $f_{\text{original}}$ on the target data $x_{\text{target}}$ until the original model overfits to the target data. Let $f_{\text{overfit}}$ denote the overfitting model. Then the difference overfitted and original model $f_{\text{overfit}} - f_{\text{original}}$ can be used as the direction towards learning $x_{\text{target}}$, and intuitively its negative direction can be used for unlearning the target data. Therefore, TV defines the unlearning model as $f_{\text{original}} - \alpha \cdot (f_{\text{overfit}} - f_{\text{original}})$. A larger value of parameter $\alpha$ gives a higher degree of unlearning but will hurt the model utility.
- *Who's harry potter* (WHP; (Eldan & Russinovich, 2023)) is based on a similar idea as TV and uses the overfitted model $f_{\text{overfit}}$. Instead of being guided by the difference in weights it uses the probability. Let $P_f(x_t|x_{1:t-1})$ denote the next token $x_t$ distribution given the language model $f$ and prompt $x_{1:t-1}$. WHP samples the next token by the token distribution defined as

$$P_{f_{\text{original}}}(x_t|x_{1:t-1}) - \alpha \max(P_{f_{\text{overfit}}}(x_t|x_{1:t-1}) - P_{f_{\text{original}}}(x_t|x_{1:t-1}), 0). \qquad (3)$$

The role of $\alpha$ is similar to $\alpha$ in TV.

**Target LLMs.** We evaluate with four popular LLMs: GPT2-XL ((Radford et al., 2019), 1.5B) Phi-1.5 ((Li et al., 2023), 1.3B), Llama2-7b ((Touvron et al., 2023), 7B), Llama3-8b ((Dubey et al., 2024), 8B). We first finetune these pre-trained LLMs on our synthetic dataset EDU-RELAT; see Appendix D for more finetuning details. After finetuning, all LLMs have 100% accuracy on the synthetic facts. The finetuned LLMs are the target to study the unlearning.

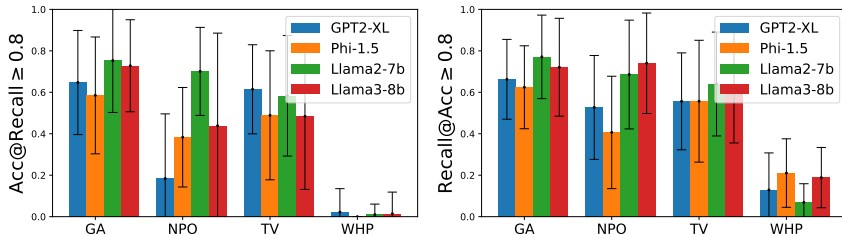

Figure 3: Acc@Recall$\geq 0.8$ and Recall@Acc$\geq 0.8$ of four unlearning methods when evaluated with four LLMs. We observe that there is no unlearning method reaching the region of both Recall$\geq 0.8$ and Accuracy$\geq 0.8$.

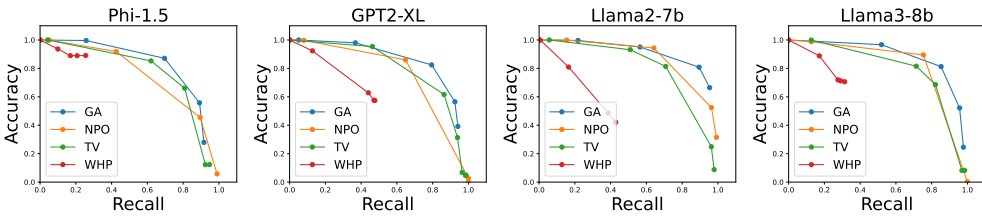

Figure 4: Acc-Recall curve when testing four methods for deeply unlearning from four LLMs.

**Target data and evaluation metric.** We have 11 different family relationships (e.g., *child*) in the synthetic knowledge base EDU-RELAT. For each family relationship, we pick 5 facts, which results in 55 facts in total for the unlearning evaluation. Our task is deep unlearning *single* fact and report the average performance among these 55 facts.

Specifically, for each fact, we run four unlearning methods from four LLMs and measure the recall (Equation1) and accuracy (Equation 2). Because each unlearning method has its parameter to control the trade-off between unlearning performance and model utility, to unlearn each target fact, we can collect a list of accuracy and recall by varying this parameter. See Appendix D for the details of the hyperparameter settings. We calculate the accuracy when the recall is larger than $0.8$ (Acc@Recall$\geq$ $0.8$; higher is better) and the recall when the accuracy is larger than $0.8$ (Acc@Recall$\geq 0.8$; higher is better), and report their average values among the 55 facts.

### 5.2 RESULTS

**Main observation: no unlearning method succeeds in deep unlearning even for just a single fact.** In Figure 3, we show the average Acc@Recall$\geq 0.8$ and Recall@Acc$\geq 0.8$ across 55 target facts for four unlearning methods on four LLMs, as well as the standard deviation presented as the error bar. It is observed that no unlearning method reaches the region of both Recall$\geq 0.8$ and Accuracy$\geq 0.8$; this means that all unlearning methods are not capable of attaining a high degree of deep unlearning while keeping an LLM utility after the unlearning. Notice that accuracy of $0.8$, i.e., dropping $0.2$ from 1, is actually a high cost, as this is the cost of unlearning only single fact; in practice, there will usually be more target facts and hence a harder setting. For easier future comparison, we also report the numbers (Acc@Recall$\geq 0.8$ and Acc@Recall$\geq 0.8$) in Table 3.

Indeed, GA is a generic unlearning method, and NPO, GA and WHP are proposed for 'concept or topic' unlearning where the target is usually a large corpus rather than single facts, otherwise the reinforced model $f_{\text{overfit}}$ may not be effective in estimating the learning direction. This mismatch of use cases may explain their performance, which motivates the design of new algorithms tailored to our deep unlearning setting.

**Comparison among four unlearning methods.** We visualize the accuracy-recall curve in Figure 4, where each point is the average accuracy and recall of an unlearning method with one hyperparameter across the evaluation of 55 target facts. We observe that WHP is less promising on all four LLMs. Particularly, its curve stops at a low recall; this is also reflected in Figure 3 where the

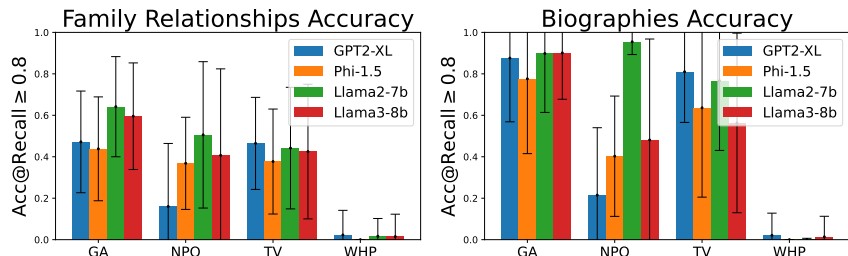

Figure 6: Acc@Recall$\geq 0.8$ on facts in family relationships and biographies. It is observed that the accuracy of biographical facts can be retained better than the accuracy of family relationships.

| Metric | Recall@Acc$\geq 0.8$ ($\uparrow$) | | | | Acc@Recall$\geq 0.8$ ($\uparrow$) | | | |
|---|---|---|---|---|---|---|---|---|
| Unlearning methods | GA | NPO | TV | WHP | GA | NPO | TV | WHP |
| GPT2-XL | 0.65 | 0.18 | **0.61** | 0.02 | 0.66 | 0.53 | 0.56 | 0.13 |
| Phi-1.5 | 0.59 | 0.38 | 0.49 | 0.00 | 0.62 | 0.41 | 0.56 | 0.21 |
| Llama2-7b | **0.75** | **0.70** | 0.58 | 0.01 | **0.77** | 0.69 | **0.64** | 0.07 |
| Llama3-8b | 0.73 | 0.44 | 0.48 | 0.01 | 0.72 | **0.74** | 0.63 | 0.19 |

Table 3: Recall@Acc$\geq 0.8$ and Acc@Recall$\geq 0.8$ of four unlearning methods on four LLMs. Relatively promising methods perform better on larger models Llama2-7b and Llama3-8b as highlighted.

Acc@Recall$\geq 0.8$ of WHP is very low. This is because of the definition as shown in Equation 3: for those negative dimensionalities in $P_{f_{\text{overfit}}}(x_t|x_{1:t-1}) - P_{f_{\text{original}}}(x_t|x_{1:t-1})$, they are invariant when varying $\alpha$ due to the operator $\max(\cdot, 0)$, even if we selected $\alpha = 10^8$ in our experiment.

**Superficial unlearning versus deep unlearning.** We measure the accuracy when the unlearning method has unlearnt the target fact but not necessarily any deep unlearning set (Acc@Superficial Unlearning). As shown in Figure 5, we find that GA is capable of carrying out this superficial unlearning – it can successfully unlearn the target fact without losing significant accuracy.

However, as discussed in the previous paragraph, even the best method GA loses more than $0.2$ in accuracy to attain recall$\geq 0.8$ of deep unlearning. This is because deep unlearning is a more challenging setting than superficial unlearning – deep unlearning single fact in EDU-RELAT can require unlearning more than 10 facts from the LLM by the definition.

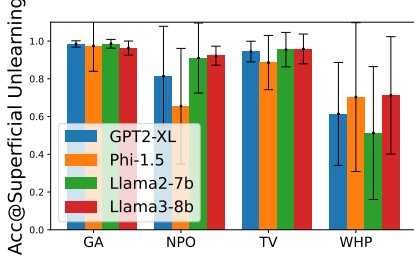

**Unlearning on small models versus unlearning on larger models.** As shown in Figure 3, while the poor-performing WHP has consistently low performance across four LLMs, relatively more effective methods GA, NPO and TV perform better on larger LLMs (Llama2-7b and Llama3-8b) than the smaller LLMs (Phi-1.5 and GPT2-XL). To be specific, the Acc@Recall$\geq 0.8$ of GA, NPO or TV on Llama2-7b or Llama3-8b is around $0.1$ or more higher than it on Phi-1.5 or GPT2-XL; It is observed

Figure 5: Accuracy@Superficial Unlearning, where only the target fact itself is required to be unlearnt. GA successfully unlearns the target fact without losing significant accuracy.

similarly for Recall@Acc$\geq 0.8$. the Recall@Acc$\geq 0.8$ of NPO on Llama2-7b or Llama3-8b is also about $0.2$ higher. We hypothesize this is because larger LLM has a better inherent understanding of the correlations between facts, which can be important to perform well in deep unlearning.

**Family relationships accuracy versus biographies accuracy.** We take a closer look at the accuracy of two subgroups of the knowledge base, the family relationships and biographies. The results are plotted in Figure 6. We observe that the accuracy of biographical facts can be retained better than the accuracy of family relationships. Specifically, the Acc@Recall$\geq 0.8$ of GA on biographical facts is higher than 0.8 on Llama2-7b and Llama3-8b, while it is only 0.6 on facts in family relationships. This is because, during unlearning, facts in family relationships are closer to the target facts (which are also family relationships) than the biographical facts and are likely to get more easily affected.

## 6 RELATED WORK

**Benchmarks and evaluations in LLM unlearning.** TOFU (Maini et al., 2024) is a benchmark containing fictitious authors and their related biographic question-answering texts such as "Awards" and "Book". They evaluate the unlearning utility by comparing the answer from LLM given the to-be-unlearned question and the ground truth answer. WMDP (Li et al., 2024) provides knowledge in biosecurity, cybersecurity, and chemical security, which matches the realistic desire for studying unlearning. A more recent benchmark MUSE (Shi et al., 2024) in the domain of news articles and books enriches the evaluation by introducing the metric from both memorization and privacy leakage aspects. Yao et al. (2024) introduces a benchmark of evaluating the unlearning in pre-trained data and the metric of unlearning utility is to compute the perplexity of the data from the memorization aspect. Patil et al. (2024) and Łucki et al. (2024) evaluate the unlearning from an adversarial attack aspect of knowledge extraction. This branch of work focuses on proposing more realistic domains and more robust ways to evaluate the unlearning, and the challenge at their benchmark is to unlearn a large batch of facts or texts while keeping the model utility. However, none of them consider the interrelation between the target facts and other facts also in the LLM. Deep unlearning in our paper is introduced to capture the connections among multiple facts in the unlearning, and we found that deep unlearning is challenging enough for just unlearning a single fact.

**Unlearning methods in LLM.** In addition to the methods evaluated in Section 5, one popular extension is assuming the existence of a "retain" set independent of the target facts. When doing gradient ascent or other gradient-based variants, Yao et al. (2023) and Chen & Yang (2023) minimize the loss on the "retain" set simultaneously to avoid quickly losing other irrelevant facts and hence help with the model utility. Another popular category is the model-editing based (Meng et al., 2022; 2023; Wang et al., 2024), which hypothesizes that the knowledge is saved in certain MLPs in the transformer and proposes an explicit-form solution for the weight update to unlearn the target facts.

**Other settings of machine unlearning.** There are other settings of machine unlearning because of different purposes and model structures. Ginart et al. (2019) and Guo et al. (2020) define the goal of unlearning as the model trained without a single person's data to meet the data regulation for data owner's rights. Tiwary et al. (2023); Kong & Chaudhuri (2024) study the concept or feature removal in the generative models, e.g. remove "white breast" in all generated bird photos; this is to avoid generating violent and racial content.

## 7 CONCLUSION AND DISCUSSION

**Conclusion.** In this paper, we propose a new setting for machine unlearning, referred to as *deep unlearning*, aimed at identifying reliable fact unlearning. As a starting point, we construct a synthetic dataset EDU-RELAT of family relationships and biographies as a benchmark for research in this emerging setting. From empirical evaluation using our metrics, we find that current unlearning methods are not capable of deeply unlearning even a single fact while keeping the model utility. We hypothesize that this shortcoming arises from these methods not fully considering the nature of facts and the deductions between each other.

**Discussion on deep unlearning real-world facts.** Current unlearning methods would likely have numerically higher accuracy on real data than on our benchmark, because the LLM knowledge base is not complete and the minimal deep unlearning set can be smaller. However, deep unlearning in real data is actually more challenging – because the unlearner would need to reason about missing facts, and decide if they might be in the knowledge base of the LLM. This is also indicated in the algorithm for finding the deep unlearning set (Algorithm 3), where the crucial step 4 involves *deductive closure of the knowledge base* $\Omega(\mathcal{K}, \mathcal{R})$ rather than the knowledge base $\mathcal{K}$ itself.

**Future work.** This work opens several promising directions for future research. Firstly, more effective methods can be developed with the deep unlearning setting, with an awareness of the connections between facts. One can assume the complete set of rules and the entire knowledge base may or may not be fully known, referred to as the white-box or the black-box scenarios. Additionally, there is potential to construct a more nuanced framework that captures more intricate facts and sophisticated deductive processes beyond the current scope of relations and logical rules. Such advancements will enhance the modeling of deep unlearning and contribute to the development of methods for deeply unlearning a broader range of facts.

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

## A  THE GURANTEEE OF ALGORITHM 3

In this section, we are going to prove that $\hat{M}_{k,\mathcal{R},\mathcal{K}}$ returned by Algorithm 3 is a collection of minimal deep unlearning sets.

*Proof.* We can first prove $k \notin \Omega(\mathcal{K}\backslash U_k, \mathcal{R})$, where $U_k$ at line 3 in Algorithm 3 is returned by Algorithm 1. The proof has two steps:

1. We can have $\Omega(\Omega(\mathcal{K}, \mathcal{R})\backslash \hat{U}_k, \mathcal{R}) = \Omega(\mathcal{K}, \mathcal{R})\backslash \hat{U}_k$, where $\hat{U}_k$ here is the $\hat{U}_k$ after line 8 in Algorithm 1. Otherwise, by the definition of deductive closure, there exists $k' \notin \Omega(\mathcal{K}, \mathcal{R})\backslash \hat{U}_k$ and $k'$ can be deduced from initiation of the rule where all facts on the left of the rule are in $\Omega(\mathcal{K}, \mathcal{R})\backslash \hat{U}_k$, i.e. not in $\hat{U}_k$. However, this can be a contradiction because if $k' \notin \Omega(\mathcal{K}, \mathcal{R})\backslash \hat{U}_k$, $k'$ must be in $\hat{U}_k$ and line 5-7 in Algorithm 1 can guarantee that for any initiation of any rule that can imply $k'$, there is at least one fact on the left of the rule in $\hat{U}_k$.

2. From line 1 in Algorithm 1, we know that $k \in \hat{U}_k$. This means that $k \notin \Omega(\mathcal{K}, \mathcal{R})\backslash \hat{U}_k = \Omega(\Omega(\mathcal{K}, \mathcal{R})\backslash \hat{U}_k, \mathcal{R})$, where the equality is from step 1. On the other hand, $(\mathcal{K}\backslash U_k) = (\mathcal{K}\backslash \hat{U}_k) \subseteq \Omega(\mathcal{K}, \mathcal{R})\backslash \hat{U}_k)$ where the equality comes from the definition $U_k = \mathcal{K} \cap \hat{U}_k$ at line 9 in Algorithm 1. $k \notin \Omega(\Omega(\mathcal{K}, \mathcal{R})\backslash \hat{U}_k, \mathcal{R})$ and $(\mathcal{K}\backslash U_k) \subseteq \Omega(\mathcal{K}, \mathcal{R})\backslash \hat{U}_k)$ together imply $k \notin \Omega(\mathcal{K}\backslash U_k, \mathcal{R})$.

We now have $k \notin \Omega(\mathcal{K}\backslash U_k, \mathcal{R})$, then we are going to prove $U_k^*$ returned by Algorithm 2 is a minimal deep unlearning set. From Algorithm 2, it is obvious that $k \notin \Omega(\mathcal{K}\backslash U_k^*, \mathcal{R})$. If it is not the minimal deep unlearning set, then there exists $U' \subset U_k^*$ s.t. $k \notin \Omega(\mathcal{K}\backslash U, \mathcal{R})$ and there is an $k'$ s.t. $k' \notin U_k^*$ and $k' \in U'$. However, this is a contradiction, because Algorithm 2 only returns $U_k^*$ if $\forall k' \notin U_k^*$, $k \in \Omega(\mathcal{K}\backslash U_k^*\backslash\{k'\}, \mathcal{R})$.

Now we can conclude $U_k^*$ at line 4 in Algorithm 3 is a minimal deep unlearning set, and our proof is done. $\qquad\square$

## B  STATISTICS OF EDU-RELAT

For a better understanding of our synthetic dataset EDU-RELAT, we present some statistics here.

- The distribution of family relations Figure 9. It is observed that *child*, *father* and *mother* are top-three relationships in our dataset.
- The distribution of the birth year is plotted in Figure 10, in a range of 1890 - 2000.
- The set of jobs, collected from the job list across years 1900-2020, is {Lawyer, Physician, Sales Manager, Machinist, Systems Administrator, Factory Worker, Police Officer, Plumber, Firefighter, Librarian, Television Repairman, Pilot, Network Administrator, Carpenter, Steelworker, Financial Analyst, Clerk, Bank Teller, Secretary, Banker, Radio Technician, Customer Service Representative, Remote Work Consultant, Postman, Baker, Movie Theater Usher, Stenographer, Software Engineer, Doctor, Maid, Construction Worker, Systems Analyst, Electrician, Auto Mechanic, Account Manager, Journalist, Welder, Mechanic, Real Estate Agent, Radio DJ, Telephone Operator, Chauffeur, Taxi Driver, Telemarketer, Car Salesman, Truck Driver, Accountant, Teacher, Airline Pilot, Draftsman, Software Developer, Nurse, Advertising Executive, Graphic Designer, IT Consultant}
- The distribution of birthplace is summarized in Figure 11.

## C  EMPIRICALLY EVALUATING ALGORITHM 3 ON EDU-RELAT

By running Algorithm 3 on the facts from our synthetic dataset, we find that Algorithm 3 does generate a rich set of minimal deep unlearning sets. In Figure 7, we show the number of minimal deep unlearning sets founded by Algorithm 3 in a histogram. It is observed that for more than half of

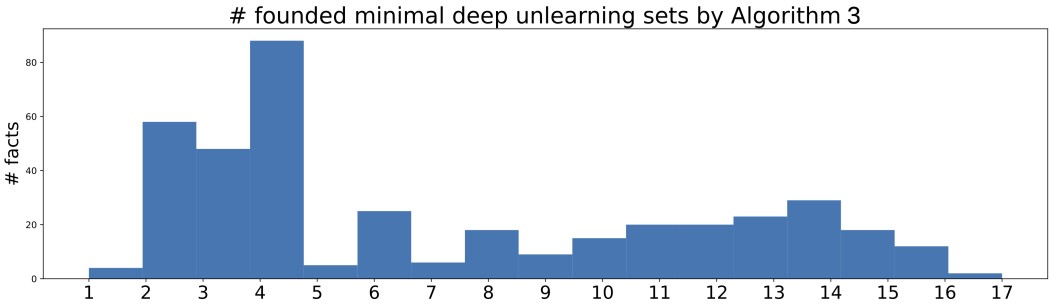

Figure 7: Histogram of # minimal deep unlearning sets founded by Algorithm 3.

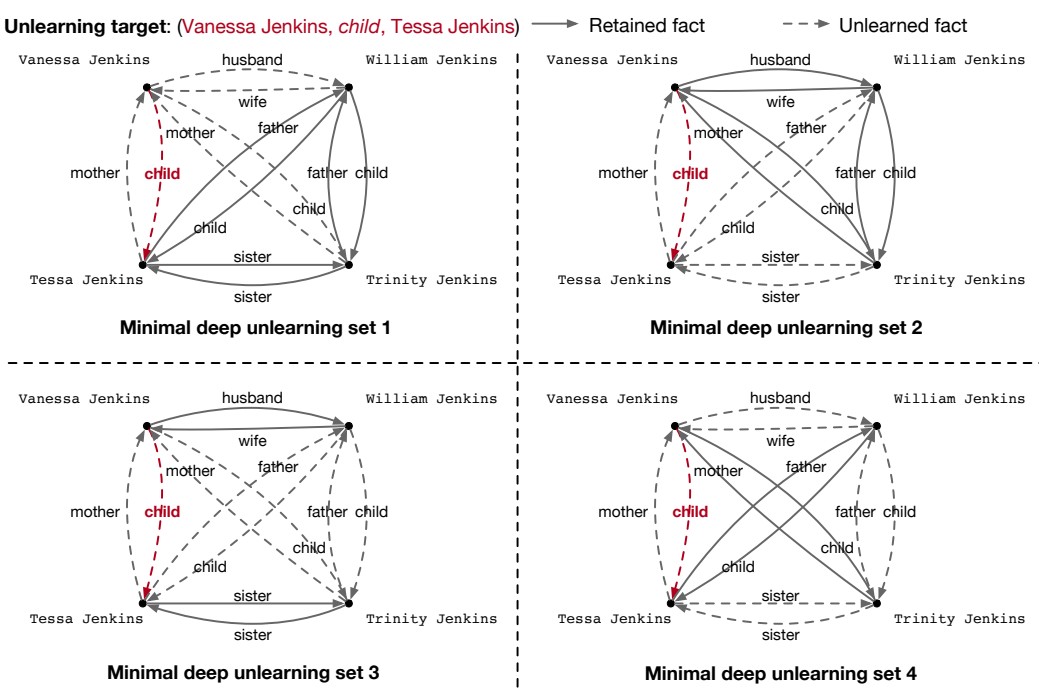

Figure 8: An example of 4 minimal deep unlearning sets founded by Algorithm 3.

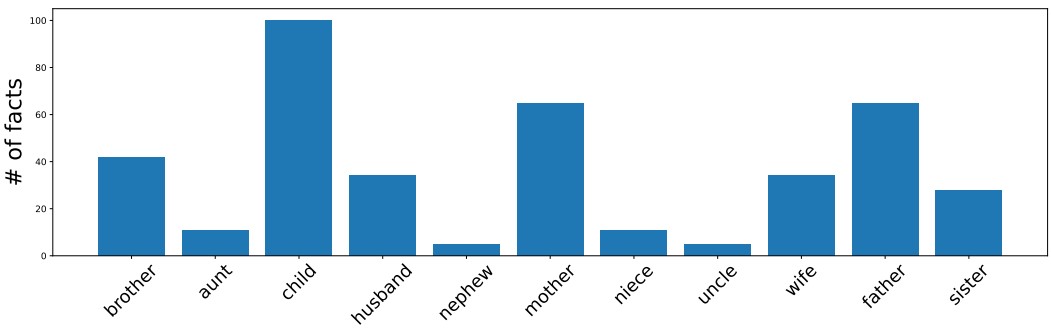

Figure 9: Distribution of relations in our synthetic dataset.

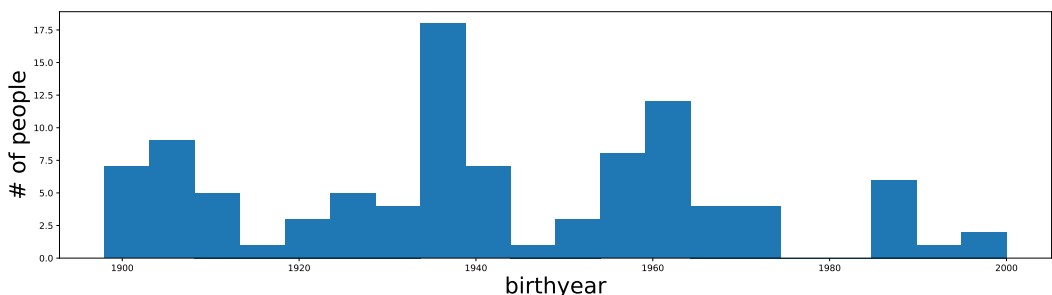

Figure 10: Distribution of birth years of fictitious people in our synthetic dataset.

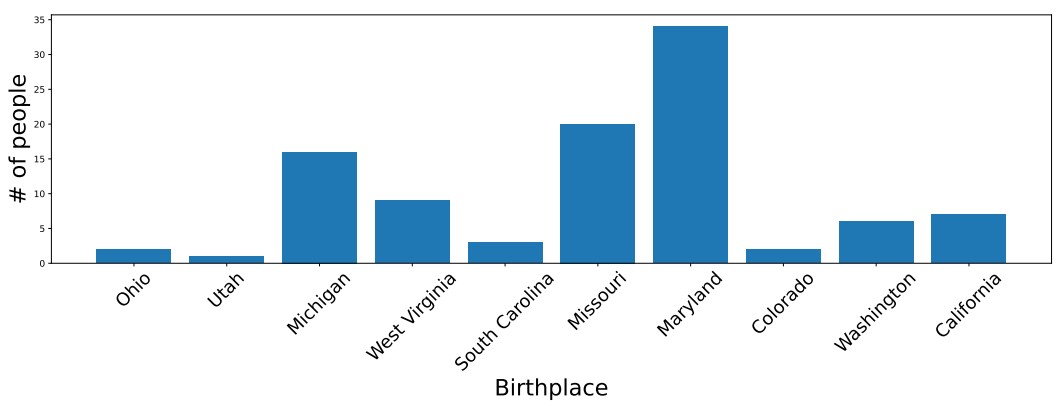

Figure 11: Distribution of birthplaces of fictitious people in our synthetic dataset.

the facts as the target fact, Algorithm 3 can return 6-17 different minimal deep unlearning sets. This demonstrates the effectiveness of Algorithm 3 and hence a good approximation when computing the recall in Equation 1.

We also show an example of minimal deep unlearning sets founded by Algorithm 3 in Figure **??**.

## D   MORE DETAILS ON EXPERIMENTAL SETTINGS

**Details of finetuning LLMs on EDU-RELAT.**    The finetuning is under the question-answering format, where the question is given in the prompt and the loss is computed from the answer. The batch size of finetuning on all four LLMs is 16. The learning rate is $2e-5$ for GPT-XL and Phi-1.5 and $1e-5$ for Llama2-7b and Llama3-8b; the learning rate scheduler is the linear scheduler from HuggingFace (Wolf, 2019). The number of epochs is 10 for Phi-1.5, Llama2-7b, and Llama3-8b and 15 GPT-XL to guarantee a full memorization after finetuning.

**Details of hyperparameters in unlearning methods**    For GA, the learning rate is $2e-5$ for GPT-XL and Phi-1.5 and $1e-5$ for Llama2-7b and Llama3-8b; the learning rate scheduler is the linear scheduler from HuggingFace (Wolf, 2019). The hyperparameter of the optimization iteration $T$ is selected from $\{1, 2, 4, 8, 16\}$ for Phi-1.5, Llama2-7b and Llama3-8b and $\{1, 2, 4, 8, 16, 32\}$ for GPT-XL. For NPO, the learning rate is $4e-5$ for GPT-XL and Phi-1.5 and $2e-5$ for Llama2-7b and Llama3-8b; the learning rate scheduler is the linear scheduler from HuggingFace (Wolf, 2019). The hyperparameter of the optimization iteration $T$ is selected from $\{1, 2, 4, 8, 16\}$ for Phi-1.5, Llama2-7b and Llama3-8b and $\{1, 2, 4, 8, 16, 32\}$ for GPT-XL. For both TV and WHP, the "overfit" model is finetuned with 10 more iterations on the target data point. In TV, the hyperparameter $\alpha$ is from

$\{0.2, 1.0, 5.0, 10.0, 30.0, 60.0, 80.0\}$ for GPT-XL and $\{0.2, 0.5, 1.0, 5.0, 10.0\}$ for Phi-1.5, Llama2-7b and Llama3-8b. In WHP, the hyperparameter $\alpha$ is from $\{10^1, 10^3, 10^5, 10^7, 10^8\}$.

