# OpenReview forum: "Evaluating Deep Unlearning in Large Language Models"
_ICLR.cc/2025/Conference — Submitted to ICLR 2025_

### Official Review · Reviewer_yUcx · 2024-11-01

**Soundness:** 3
**Presentation:** 3
**Contribution:** 2
**Rating:** 5
**Confidence:** 3

**Summary:**

This paper introduces the concept of "deep unlearning", which studies the unlearning tasks in large language models where logical relationships between facts need to be considered. The authors constructed a synthetic dataset EDU-RELAT, containing family relationships and biographical information, to evaluate four unlearning methods across four different-sized LLMs.

**Strengths:**

1. Presents an important and novel problem - deep unlearning that considers logical reasoning between facts
2. Constructs a structured evaluation dataset with reasonable logical rules and relationships
3. Proposes reasonable evaluation metrics (recall and accuracy) and designs approximate algorithms to compute these metrics
4. Comprehensive experiments that reveal the limitations of current methods in deep unlearning

**Weaknesses:**

1. While deep unlearning sounds reasonable overall, the paper's setting may not align with practical scenarios. In real machine unlearning cases, related knowledge is typically forgotten together (e.g., forgetting all content related to Harry Potter) rather than just forgetting a single relationship. Under this setting, is it still important to forget all knowledge that could potentially derive the current relationship?

2. Given that real-world relationships can be far more complex than the R defined in this paper's dataset, could there be situations where forgetting one piece of knowledge requires forgetting an excessive amount of content? In multi-hop reasoning scenarios, a large amount of knowledge might need to be forgotten, which raises another question: is deep unlearning always necessary in unlearning scenarios? If unlearning is applied for copyright protection, can knowledge derived through multi-hop reasoning constitute infringement?

3. The paper only focuses on logical rules in the specific domain of family relationships, which is rather narrow in scope.

4. No new solutions are proposed to improve deep unlearning effectiveness, remaining only at the problem analysis level.

**Questions:**

Please refer to the weakness part.

---

> ### Author Response · Authors · 2024-11-21
>
> Thank you for your positive comments (novel and important problem, structured dataset, reasonable metrics and comprehensive experiments) and your constructive feedback. We reply to your concerns below.
>
> **Practical scenarios of (single) fact unlearning (Reply to Weakness 1&2)**: Thank you for raising your confusion. Machine unlearning indeed has many use cases, such as concept removal and copyright protection, and in different use cases, the definition and corresponding methodology can be tailored differently. Our (single) fact unlearning is particularly important to address privacy risks. A realistic scenario can be: for celebrities, we want to unlearn their home addresses, but keep their public profiles such as the rewards and achievements. In the revised introduction (line 34-46 of the revised pdf), we have clarified the difference between fact unlearning and other variants of unlearning such as data removal and copyright protection and have introduced the detailed use case of fact unlearning.
>
> **The study of solutions (Reply to Weakness 4)**: We would like to thank you for expecting the study of new solutions. We agree that this is an important direction of future work and have discussed this in the last section. In this paper, we believe the problem definition, framework and evaluation themselves are already important. In addition, the solution study and problem proposal can have conflict of interest in one paper, in the sense that the problem proposal can potentially be flavored to the proposed solution – that’s why we consciously made this decision.

---

> ### Author Response · Authors · 2024-12-03
>
> Dear reviewer yUcx,
>
> Thank you for your constructive feedback. As the discussion stage is closing in a day, we would appreciate it if you could take a look to our responses and let us know if your questions have been addressed. We are happy to discuss if there are any additional questions. Thank you for your time!
>
> Best regards,
>
> Authors

---

### Official Review · Reviewer_nGBY · 2024-11-03

**Soundness:** 2
**Presentation:** 1
**Contribution:** 2
**Rating:** 5
**Confidence:** 5

**Summary:**

This paper introduces "deep unlearning" as a novel approach in the domain of large language models (LLMs), emphasizing its importance in effectively erasing certain facts. As the target fact can be deduced from logical rules, superficial unlearning methods, which solely unlearn the target fact, cannot unlearn it successfully. The authors also present new metrics, including Accuracy and Recall, to evaluate this process, backed by experiments across various LLMs and unlearning methods.

**Strengths:**

1. Novel Contribution: The introduction of deep unlearning and a curated dataset highlights a significant gap in current research, drawing attention to an important and underexplored issue.
2. Innovative Metrics: The proposal of Recall as a new evaluation metric, accompanied by a detailed algorithm for addressing the NP-hard nature of the problem, demonstrates thoughtful consideration of the challenges involved in deep unlearning.
3. Comprehensive Experiments: The thorough experimental setup across four LLMs and unlearning methods strengthens the paper's credibility and provides valuable insights into the effectiveness of the proposed approach.

**Weaknesses:**

1. Code Availability: The lack of code and dataset release until after acceptance may hinder reproducibility and limit the community's ability to validate the findings. If possible, we encourage the author to give an anonymous link to the git repo. If it is forbidden in rebuttal process, you can include more raw samples in Appendix with explanation. More importantly, as you proposed a benchmark, the statistics of the datasets are of vital importance, which should be discussed in the main part, rather than appendix.
2. Missing Results: The author did not report the accuracy on the dataset of the models fine-tuned but not subjected to unlearning, which makes it difficult to gauge the impact of the unlearning methods accurately. Please include the baseline accuracy results for the fine-tuned models before unlearning.
3. Poor Presentation: The author should improve the writing and figure for better illustration, especially the figures. For instance, Figure 3 and 6 are not clear to the analyzed conclusions. Maybe a organized table will be better. From my perspective, for this work, the related work is crucial for the understanding of the benchmark, which can be moved to Sec 2.

**Questions:**

Refer to peaknesses, I hope that you can solve these concerns.

**Details Of Ethics Concerns:**

Refer to Weaknesses.

---

> ### Author Response · Authors · 2024-11-21
>
> Thank you for your positive comments (novel contributions, innovative metrics and comprehensive experiments) and constructive feedback. We reply to your concerns below.
>
> **Code availability (Reply to Weakness 1)**: Thank you for your suggestion! We would like to provide our anonymous link (https://anonymous.4open.science/r/deep_unlearning_anonymous-2C73) including the code of all evaluated methods and the dataset. We also have put this link in the revision  (as shown in the abstract and experiment section).
>
> **Results of finetuned models (Reply to Weakness 2)**: Thanks for pointing this out. After finetuning the models on our synthetic data, the accuracy is 100% for every finetuned model. We have added this description in line 377 of the revised pdf.
>
> **Organization of the result presentation and related work (Reply to Weakness 3)**: Thank you for your suggestion, which helped us to improve the presentation. The results of Figure 3 have been also presented in Table 3. We further highlighted the best value across models in the revision. We agree that having related work before introducing the details of our new problem can help with the understanding. We will move the related work section to section 2 in our final revision.

---

> > ### Comment · Reviewer_nGBY · 2024-11-27
> >
> > Thank you for the clarification. I have no further question, however I would like to keep my score.

---

> ### Author Response · Authors · 2024-11-27
>
> Dear Reviewer nGBY,
>
> Thank you for your reply. We would really appreciate it if you could let us know why you would like to keep your score, so we could improve our paper further!
>
> Best Regards,
>
> Authors

---

### Official Review · Reviewer_wbf1 · 2024-11-04

**Soundness:** 2
**Presentation:** 3
**Contribution:** 2
**Rating:** 6
**Confidence:** 4

**Summary:**

This paper presents a critique of current unlearning methods through the lens of fact unlearning - it shows that while current unlearning methods can unlearn the target fact in isolation, they often fail to unlearn other related facts through which the target fact can be logically deduced, thus negating the effect of unlearning. The authors refer to the task of unlearning additional facts which are logically related to the target fact, as "deep unlearning". They introduce 2 metrics based on set overlap: recall and accuracy, which quantify the extent of "deep unlearning". All experiments are conducted on a small, synthetic dataset derived from a synthetic knowledge base and a set of logical rules.

**Strengths:**

This paper extends the discussion on the "illusion of machine unlearning" vis-à-vis current unlearning methods, to the relatively simple yet seemingly hard task of unlearning a single fact from the LLM's parametric memory, given that logically related facts are also present in the LLM. This work shows, somewhat surprisingly, that when current unlearning methods are tasked with erasing a single fact from the LLMs parametric knowledge, they do not erase related facts which also share entities/objects with the target fact, for e.g. erasing the target fact **F1**: Y is child of X, may fail to erase logically related facts such as: **F2**: Z is husband of X, and **F3**: Z is father of Y, even though Y as an entity appears in F3, and this would allow the erased fact to re-surface via logical deduction. On a synthetic toy dataset, they provide a reasonable set overlap based metric for quantifying the degree of such unlearning. While it is already known that most knowledge that is supposedly deleted by current unlearning methods can be re-surfaced via adversarial/jailbreak prompting and probing, this work highlights limitations of current methods at the atomic level of single-fact deletion, hinting at a tension between an LLM's reasoning capabilities and the effectiveness of unlearning techniques.

**Weaknesses:**

- Firstly, I find it a bit counter-intuitive that even an unlearning method such as WHP (Who's Harry Potter) would not suppress the effect of other facts which share entities with the target fact, such as in the case of facts related to familial connections where a target entity appears verbatim in other facts. This leads me to wonder if the failure to "deeply unlearn" a target fact is simply an artifact of the way in which the unlearning methods are applied e.g. WHP requires finetuning a reinforced model on a corpus that is much larger than just a handful of facts, in order for the logit difference with the baseline to be significant. The paper does not really discuss how each of the unlearning methods are actually applied, and whether any of the tested unlearning methods are even compatible with the setting of single fact unlearning - in my view, methods such as NPO, TV, and WHP are clearly more suited for a setting where the unlearning target is defined around a *concept* or *topic*, rather than single **fact**. A discussion around the compatibility of the evaluated unlearning methods with the single fact unlearning setting, and what role the small size of the synthetic dataset plays in limiting the effectiveness of unlearning methods, would be a useful addition. For example, if the WHP method is properly applied as it is designed i.e. for concept unlearning rather than fact unlearning, I imagine that it would successfully suppress the fact that "Harry is child of Lily" even if it was fine-tuned on other facts that mentioned that "James is father of Harry" and "James is husband of Lily". To help readers better understand the experimental setup and interpret the results, I recommend that the authors discuss the adaptations they made to apply WHP, NPO, TV etc., to the fact unlearning task.

- The synthetic dataset appears to be biased towards highly deducible relationships (e.g., familial connections) which is not really representative of real-world knowledge structures where logical connections are typically more complex and less *deducible*. In my view, the authors should have included an evaluation on a dataset extracted from a real-world knowledge base with partial/noisy/incomplete logical connections between facts - my guess is that since most real facts require more than just simple deductive reasoning e.g. may require multi-hop reasoning, this problem of knowledge deductibility or fact reconstruction would be less of an issue even with current unlearning methods. While the authors claim that it is hard to conduct prompt based evaluations for determining if a fact is in the LM (false negatives), I'd like to point out that alternatives to the simple prompts used by the authors (as shown in Table 1) do exist (see [1]), such as MCQ-based binary choice Q&A or latent supervised/unsupervised probing. It would also useful have been to see how the "approximate" recall and accuracy metrics will scale to real KBs where many of the logical connections between facts are unknown.

- The authors should discuss the limitations of their current dataset and how these might affect the generalizability of the results.

- Minor typos: typo in appendix C, should be “algorithm 3” instead of “Algorithm 9”.  Caption of Figure 7 should also say “algorithm 3” instead of “algorithm 1”.  Typos in each caption of subfigures in figure 8: should say “Minimal Deep Unlearning” sets.

[1] [Eight Methods to Evaluate Robust Unlearning in LLMs](https://arxiv.org/abs/2402.16835)

**Questions:**

Q1: How would the accuracy and recall metrics change w.r.t. to the # of minimal deep learning sets discovered when applied on a KB with incomplete/partial logical dependencies? Would recall increase since the minimal deep unlearning set is smaller and closer in size to the actual set unlearned by the algorithm, thus having higher overlap? What about accuracy?

Q2: Given the small size of the target facts to be unlearnt, how do you satisfy the fine-tuning data requirements of methods such as NPO & WHP?

Q3: What motivated 0.8 as threshold value for comparing recall/accuracy (Figures 3 and 6)?

---

> ### Author Response · Authors · 2024-11-21
>
> Thank you for your insightful comment about the WHP and real-world knowledge base! Please see our reply to your questions and concerns below.
>
> **The adaptation of WHP (Reply to Weakness 1 and Question 2)**: Thank you for raising this point. We agree that WHP is more suited for *concept or topic* unlearning, and there is a mismatch in use cases – the use-case in this work is to unlearn the (single) fact. We have revised our intro (line 39-47 in the revision) to highlight the different use cases of unlearning in LLMs and added the insight of WHP provided by your review (line 424-428 in the revision) to emphasize WHP is more suited to concept or topic unlearning where the unlearning data is usually a large corpus. This further *motivates that for the use case of (single) fact deep unlearning, different and newer methods may need to be designed*.
>
> For the experimental setup of WHP, we tried our best to adapt it to the task of single fact unlearning. As described in Appendix D, the reinforced model is finetuned with only the fact to be unlearnt. For reproducibility, we also attach this anonymous link (https://anonymous.4open.science/r/deep_unlearning_anonymous-2C73) including the cleaned code of all evaluated methods and the dataset in the revision (as shown in the abstract and experiment section).
>
> When we cleaned the code, we found an elusive bug in the Recall and Accuracy calculation code for TV and WHP specifically. After removing the bug and rerunning the code, we find the performance of TV looks better and that of WHP remains similar. We have updated the results in the revision and also released the cleaned code (in the link above) accordingly.
>
> **Why do we evaluate deep unlearning with synthetic facts rather than real-world facts? (Reply to Weakness 2)** We use synthetic data in order to *control* the evaluation experiments, removing potential factors for noisy evaluation such as (1) a partial observation of the underlying knowledge base in the LLM leading to a false sense of success and (2) different underlying knowledge bases across LLMs making it harder to draw consistent conclusions. For more details on these factors, kindly see line 287-301 of the revised paper. Therefore, to have better control in the evaluation, we decided to create the synthetic dataset, which is the popular option in other evaluation work as well [1, 2].
>
> Thank you for pointing out a reference for how to determine if a fact is in the LLM. This can help alleviate the factor (1), but the evaluation noise can still exist because the public knowledge base itself is incomplete.
>
> **How would current unlearning methods perform on real-world fact unlearning? (Reply to Weakness 3 and Question 1)** Current unlearning methods would likely have numerically higher accuracy on real data (which can have an incomplete LLM knowledge base) than on our benchmark. However, the problem of deep unlearning in real data is actually more challenging – because the unlearner would need to reason about missing facts and their probabilities, and decide if they might be in the knowledge base of the LLM. We have added this discussion in the revised pdf in Section 7.
>
> **The choice of threshold (Reply to Question 3)**: We choose the threshold of $0.8$ as an indication of relatively high value of accuracy and recall. We have also shown the curve of accuracy and recall in Figure 4.
>
> **Typos (Reply to Weakness 4)**: Thank you for pointing them out and thank your effort for carefully checking our Appendix too! We have corrected them accordingly in the revision.
>
> [1]Maini, Pratyush, et al. "Tofu: A task of fictitious unlearning for llms." arXiv preprint arXiv:2401.06121 (2024).
>
> [2] Allen-Zhu, Zeyuan and Li, Yuanzhi. ``Physics of Language Models: Part 3.1, Knowledge Storage and Extraction.Forty-first International Conference on Machine Learning (2024)

---

> > ### Comment · Reviewer_wbf1 · 2024-11-26
> >
> > Thanks to the authors for revising the paper to incorporate the suggested discussions. I have updated my score.

---

> > > ### Author Response · Authors · 2024-11-26
> > >
> > > Dear Reviewer wbf1,
> > >
> > > Thank you so much for increasing your recommendation!
> > >
> > > Best Regards,
> > >
> > > Authors

---

### Author Response · Authors · 2024-11-21
**General Comment by Authors**

We thank all the reviewers for their time and insightful feedback. We appreciate that the reviewers (nGBY, yUcx) find that our proposed problem is novel and important, the evaluation metric is innovative and reasonable, and the experiments are comprehensive.

For each reviewer's valuable suggestions and critiques, we respond in our individual responses and have edited some paragraphs in the revised paper to improve the paper according to the feedback; the differences are highlighted in blue. We sincerely hope to continue this insightful discussion during the discussion period and would like to thank you again for your constructive feedback!

---

### Author Response · Authors · 2024-11-25

Dear reviewers,

Thank you for your time and feedback, and we would be happy to answer further questions if there are still any concerns. Please let us know if any additional clarifications we can provide!

Best Regards,

Authors

---

### Meta-Review · Area_Chair_6Bzd · 2024-12-17

**Metareview:**

This paper introduces "deep unlearning" as a novel approach in the domain of large language models (LLMs), emphasizing its importance in effectively erasing certain facts. As the target fact can be deduced from logical rules, superficial unlearning methods, which solely unlearn the target fact, cannot unlearn it successfully. This work presents an important and novel problem - deep unlearning that considers logical reasoning between facts. The authors also present new metrics, including Accuracy and Recall, to evaluate this process, backed by experiments across various LLMs and unlearning methods.

There were several non-trivial issues raised in the reviews. First, the experiments did not compare the utility performance of a vanilla model with the unlearned model. Second, there have been concerns regarding the limited studied scope that was merely focusing on logical rules. Third, there were also concerns among the reviews regarding the quality of presentation. Some (not all) of the issues were incorporated into the revision.

**Additional Comments On Reviewer Discussion:**

ome (not all) of the issues were incorporated into the revision.

---

### Decision · Program_Chairs · 2025-01-22

Reject